# Acute Vagus Nerve Stimulation Facilitates Short Term Memory and Cognitive Flexibility in Rats

**DOI:** 10.3390/brainsci12091137

**Published:** 2022-08-26

**Authors:** Christopher M. Driskill, Jessica E. Childs, Bemisal Itmer, Jai S. Rajput, Sven Kroener

**Affiliations:** School of Behavioral and Brain Sciences, The University of Texas at Dallas, 800 West Campbell Rd., Richardson, TX 75080, USA

**Keywords:** vagus nerve stimulation, short term memory, cognitive flexibility

## Abstract

Vagus nerve stimulation (VNS) causes the release of several neuromodulators, leading to cortical activation and deactivation. The resulting preparatory cortical plasticity can be used to increase learning and memory in both rats and humans. The effects of VNS on cognition have mostly been studied either in animal models of different pathologies, and/or after extended VNS. Considerably less is known about the effects of acute VNS. Here, we examined the effects of acute VNS on short-term memory and cognitive flexibility in naïve rats, using three cognitive tasks that require comparatively brief (single session) training periods. In all tasks, VNS was delivered immediately before or during the testing phase. We used a rule-shifting task to test cognitive flexibility, a novel object recognition task to measure short-term object memory, and a delayed spontaneous alternation task to measure spatial short-term memory. We also analyzed exploratory behavior in an elevated plus maze to determine the effects of acute VNS on anxiety. Our results indicate that acute VNS can improve memory and cognitive flexibility relative to Sham-stimulation, and these effects are independent of unspecific VNS-induced changes in locomotion or anxiety.

## 1. Introduction

Vagus nerve stimulation (VNS) is used for the treatment of several neurological disorders [1,2,3]. Vagal afferents may function as an endogenous mediator of certain cognitive functions [4,5]. Ascending fibers of the vagus nerve innervate the nucleus of the solitary tract, which then relays signals to areas in the brain stem and the forebrain, including areas in the thalamus, amygdala, and hippocampus that are involved in learning and memory [6]. VNS-induced activation of brainstem nuclei causes the release of several neuromodulators, including norepinephrine (NE), acetylcholine, and serotonin [7,8,9,10,11,12,13,14], resulting in widespread cortical and subcortical activation and deactivation [15,16,17,18,19]. A number of studies have shown that pairing VNS with discrete stimuli or behaviors promotes cortical plasticity which can facilitate learning and memory in both rats [17,20,21,22,23,24,25] and humans [26,27]. The majority of previous studies that examined the effect of VNS on cognition have done so in the context of animal models of various pathologies, and/or after chronic VNS. Here, we examined the effects of acute VNS on short-term memory and cognitive flexibility in naïve rats. We used a series of behavioral tasks that require little or no training, and we limited delivery of VNS to the testing phase of each task in order to minimize the duration of VNS. These tasks included a rule-shifting task that tests cognitive flexibility (as a measure of reasoning and problem solving), a novel object recognition task, which measures (short-term) memory for objects, and delayed spontaneous alternation in a T-maze to measure spatial short-term memory. We found that acute VNS improved performance across all measures of memory and cognitive flexibility compared to Sham-stimulation. These effects were independent of potential VNS-induced reductions in anxiety, as acute VNS did not significantly affect exploratory behavior in an elevated plus maze. Our findings add to a growing body of literature that shows that even relatively brief VNS can affect cognitive functions [20,21,22,26,27,28]. In addition, our work identifies tasks covering multiple cognitive domains, that can be learned by rats in one or few sessions, and which are modulated by VNS, thus facilitating future studies into the effects of VNS on cognition.

## 2. Materials and Methods

### 2.1. Animals and Surgical Procedures

All procedures were carried out in accordance with the NIH Guide for the Care and Use of Laboratory Animals, and were approved by the Institutional Animal Care and Use Committee of The University of Texas at Dallas. A total of 65 male Sprague-Dawley rats (Taconic, Rensselaer, NY, USA) was used in these experiments. Forty-seven rats were used for the three cognitive tasks (Cross-maze Rule-shifting task, Delayed Spontaneous Alternation, Novel Object Recognition, see below), with the majority of the rats participating in two of the three tasks. In order to avoid confounds of previous experiences with behavioral tasks and/or VNS, a separate cohort of 18 rats was used to assess anxiety in the Elevated-Plus Maze. All rats were socially housed on a 12 h reverse light/dark cycle (lights off at 6:00 a.m.) with access to food and water ad libitum. The rats weighed 300–350 g (~75 days postnatal) at the start of the experiments. Custom-made cuffs for vagus nerve stimulation were implanted as described elsewhere [23,29,30]. In brief, rats were anesthetized with ketamine (85 mg/kg) and xylazine (5 mg/kg), and pre-treated with atropine (1 mg/kg) and the local anesthetic Marcaine. For VNS delivery, a stimulation input site was constructed using a head-fixed two-channel strip connector (Omnetics, Minneapolis, MN, USA) connected to leads from a custom nerve cuff platinum-iridium wire electrode in micro-renathane (0.04″ ID, 0.08″ OD, 4 mm long). The nerve cuff was placed around the left vagus nerve by accessing the vagus nerve via an incision in the ventral midcervical region of the neck. Cuff function was assessed during surgery by applying VNS (0.2 mA, 60 Hz, 10 s) and observing brief cessation of breathing due to recruitment of the Hering–Breuer reflex in anesthetized rats [24,31]. In Sham-stimulated animals, cuffs were designed to short at the level of the headstage. Sham-stimulated rats showed no evidence of the inflation reflex. After surgery rats were allowed to recover for one week. Prior to all behavioral tasks animals were handled each day for two weeks in their home cages. In addition, three days before the experiments, rats were also handled in the room where the experiments were performed and were habituated to the tether and the potential novel sensation of VNS by tethering them to a stimulator and issuing a 30 s stimulation (0.4 mA, 30 Hz, 500 us pulse width) every three minutes for 15 min. On the day of testing, animals were transferred to the behavioral room at least 30 min before testing began. Experimenters were blind to the treatment of experimental animals throughout testing and analysis. During all behavioral experiments, rats were tethered to an AM Systems stimulator. Rats in the VNS groups received a 30 s stimulation (0.4 mA, 30 Hz, 500 us pulse width) every three minutes. These stimulation parameters can modulate learning and synaptic plasticity without disrupting ongoing behavior [23,24,29]. Sham animals were identically tethered but not stimulated. 

### 2.2. Cross-Maze Rule-Shifting Task (CMRST)

In order to measure the effect of VNS on attention and cognitive flexibility, rats were trained on a rule shifting task that requires decision-making capabilities and the ability to inhibit a prepotent but inappropriate response. Procedures followed those previously described [32,33]. Briefly, rats were habituated to the maze for three days, during which all arms were baited with food reward pellets and animals were allowed to freely explore the maze. Rats were connected to stimulation cables at their head caps throughout habituation and testing, with actual stimulation only delivered on the last day (Shift-to-Visual-Cue Day). Rats cleared habituation after consuming all the pellets on the third day in less than 15 min, but usually this took less than 2 min. After habituation, three days of testing occurred with the plus maze converted into a T-maze. On the first day of testing, the Rats’ turn bias was determined. A black and white striped visual cue was placed near the entrance to one of the entry arms in a pseudorandom manner and rats were placed in the stem arm and allowed to turn left or right to obtain a food pellet. After the rat consumed the reward, it was returned to the stem arm and allowed to make another choice. If the rat chose the same arm as on the initial choice, it was returned to the stem arm until it chose the other arm and consumed the food pellet. A total of seven trails were run, and the most frequently selected arm indicated the turn bias. On the second day of testing, rats were trained on a response discrimination task that required them to learn the rule ‘always turn this way’ (left or right, direction opposite turn bias) to obtain a food pellet. The location of the stem arm was rotated among three arms (East, West and South; North unused) to prevent rats from using spatial strategies. Again, the black and white striped visual cue was placed near the entrance to one of the entry arms. Placement of this cue varied pseudorandomly to balance the cue presentation in the left or right arm over blocks of 12 trials. The order of the stem arms was similarly alternated across blocks of 12 trials. Successful training on Response Discrimination was marked when the rat made 9 correct choices in any block of 10 trials and then successfully passed a probe trial. In the probe trial, the previously unused North arm was used as the stem arm to ensure that rats were indeed following the visual cue and did not rely on an allocentric response strategy. If the probe trail was not passed, training continued until the rat made another five consecutive correct choices, at which point another probe trial was administered. On the third and final day (Shift-to-Visual-Cue day), rats were tested on their ability to shift their strategy. Instead of following the “turn this way” rule they were now required to learn to “follow the visual cue” in order to obtain food rewards. The location of the visual cue and the position of the start arm were again varied. The training and response criteria for the Shift-to-Visual-Cue Discrimination were identical to those during Response Discrimination. VNS or Sham-stimulation was delivered during Shift-to-Visual-Cue training for 30 s every three minutes. On the third day errors were scored as entries into arms that did not contain the visual cue, and they were further broken down into three subtypes to determine whether VNS altered the ability to either shift from the previously learned strategy (*perseverative errors*), or to maintain the new strategy after perseveration had ceased (*regressive errors*, or *never-reinforced errors*). In order to detect shifts in the strategies that rats used, we separated trials into blocks of four trials each. A *perseverative error* occurred when a rat made the same egocentric response as required during the Response Discrimination, but which was opposite to the direction of the arm containing the visual cue. Six of every 12 trials required the rat to respond in this fashion against the previous egocentric response strategy. A perseverative error was scored when the rat entered the incorrect arm on three or more trials per block of four trials. Once the rat made less than three perseverative errors in a block, all subsequent errors of the same type were now scored as *regressive errors*, because at this point the subject had adopted an alternative strategy at least half of the time. Finally, *never-reinforced errors* were scored when a rat entered the incorrect opposite arm on trials where the visual cue was placed on the same side congruent with the previous egocentric response strategy. 

### 2.3. Delayed Spontaneous Alternation

We used the innate tendency of rodents to alternate entries into the arms of a maze as another test of short-term memory [33]. Therefore, rats were placed into the stem arm of a T-maze and rats were left to choose between the left and the right open goal arm. Once a rat had entered an arm it was confined to this arm by blocking off the entrance of the arm. After 30 s, the barrier was removed and the animal was placed back into the stem arm and allowed to once more choose between the left and right arm. Each animal completed a total of six trials (with 2 choice runs per trial, separated by ~35 s), each separated by a 15-min interval during which rats received 30 s of VNS or Sham stimulation every three minutes. The maze was cleaned with 70% ethanol after every single run. The turn direction was recorded for each trial and the percentage of alternations between left and right arm entries within trials was calculated. Higher rates of alternation were considered indicative of enhanced short-term memory. In addition, we also analyzed alternations across the 15 min intertrial interval in an effort to assess potential changes in memory on a longer time scale.

### 2.4. Novel Object Recognition

We used a novel object recognition task to assess the effect of VNS on short-term and declarative memory as previously described [32,33]. Experiments were conducted in a 60 × 60 × 40 cm open field with white walls and a black floor. Three objects of comparable size, height, weight, and luster were used as either the familiar or novel object. All the objects used on the test were previously tested for preference by a control group of naïve rats to ensure that rats showed no inherent preference for any one of these objects. The object assigned as novel was rotated across animals to further eliminate any object specific preferences unrelated to novelty. Prior to testing, animals were handled (3 min) and habituated to the open field (10 min) for three days. During habituation, rats were exposed to two (familiar) objects. On test day, rats were placed in the open field for a one-minute habituation period, followed by a three minute ‘familiar phase’ in which animals were allowed to explore the two familiar objects. Then animals were placed in a holding cage for a 15-min retention phase where VNS or Sham-stimulation was delivered for 30 s every three minutes. The objects were cleaned with 20% ethanol and the chamber was cleaned with 70% ethanol between all trials. The configuration of the objects during each trial was changed for each animal. After the retention phase, one of the familiar objects was replaced with a novel object and rats were allowed to explore the field for three minutes. Activity during the novel phase was video-recorded and analyzed to determine the total exploration time for both the novel object and familiar object. Climbing onto an object was not considered exploratory behavior. A recognition index was determined by dividing the amount of time spent with the novel object over the total time spent investigating both objects [34]. Increased exploration of the novel object was considered as an indicator of improved short-term memory.

### 2.5. Elevated plus Maze

We used an elevated plus maze (EPM) to assess anxiety-related behavior. For this task we used a separate cohort of rats that had not participated in any of the other tests and thus had not received any VNS prior to the test. The EPM consisted of two opposing arms enclosed by 12 inches high walls, and two open arms (all arms were each 24 inches long and 4 inches wide) positioned 18 inches from the floor. Prior to the EPM test, rats were habituated to the room and to being connected to the VNS tether over three days. On the days of the test, rats received three trains of VNS (30 Hz, at 0.4 mA; 500 μs pulse width) or Sham-stimulation for 30 s in 3 min intervals before they were placed in the maze. Rats were placed at the center of the maze and their behavior was observed for five minutes. Trials were video recorded and analyzed for the time spent in the open arms, time spent in the closed arms, and the total number of entries into the open arms. 

### 2.6. Data Analysis

We used a two-way ANOVA for the Cross-maze Rule Shifting task, followed by Bonferroni-corrected *t*-tests. All other tasks were analyzed using unpaired *t*-tests (Graphpad Prism 7). Data in all figures is displayed as the mean average, with error bars representing the standard error of the mean. *p* values less than 0.05 were considered significant, and points of significance are indicated with *.

## 3. Results

We trained rats on three behavioral tests to study the effect of acute VNS on short term memory and cognitive flexibility.

### 3.1. Cross-Maze Rule Shifting Task

We used the CMRST to determine whether acute VNS can modulate cognitive flexibility. After habituation, rats (Sham, *n* = 14; VNS, *n* = 14) learned a Response Discrimination on Day 1 and then had to shift to a different strategy on Day 2 to obtain food rewards (Shift-to-Visual Cue-Discrimination). VNS and Sham-stimulation were applied non-contingently at regular 3 min intervals only on Day 2 during the Shift-to-Visual-Cue discrimination. The duration of the task varied with the rats’ performance, but no subject took longer than 2 h to reach criterion, thus capping the maximal number of VNS stimulations at 40. A two-way ANOVA with the factors test-day and treatment found a significant difference between days (F_(1,26)_ = 18.91, *p* = 0.0002) and treatment (F_(1,26)_ = 7.101, *p* = 0.0131), as well as a trend for an interaction between test-day and treatment (F_(1,26)_ = 4.031, *p* = 0.055). Bonferroni post-hoc testing showed no differences between Sham and VNS animals in the acquisition of the original strategy on Day 1 Response Day (*p* > 0.9999). In contrast, rats treated with VNS required fewer trials to shift to the new strategy on Day 2 compared to Sham animals (*p* = 0.0033; Figure 1). We further analyzed differences in the total number, as well as the types of errors (perseverative, regressive, and never reinforced) committed. Overall, VNS animals made fewer errors (unpaired t-test t_(26)_ = 2.513, *p* = 0.0185). While there was no significant effect of VNS on perseverative errors (t_(26)_ = 0.6745, *p* = 0.5060), VNS caused a significant reduction in both regressive (t_(26)_ = 2.520, *p* = 0.0182) and never reinforced errors (t_(26)_ = 2.542, *p* = 0.0173; Figure 1). Taken together, these results indicate that acute VNS enhanced cognitive flexibility and facilitated the shift to a new strategy.

### 3.2. Delayed Spontaneous Alternation

As a first test of whether acute VNS can affect short term memory function we used a delayed spontaneous alternation task [33]. We measured the frequency of spontaneous alternations between entries into the left or right goal-arm over 12 free-choice trials in a T-maze in Sham- (*n* = 14) and VNS-treated rats (*n* = 12). VNS-treated rats alternated their entries more frequently on directly subsequent trials (within-trial alternations) than Sham-stimulated animals (unpaired ttest t_(24)_ = 2.513, *p* = 0.0191; Figure 2B), indicating that VNS improved spatial short-term memory. However, VNS- and Sham-stimulated rats did not differ in their alternations when compared across trails (between-trial alternations; t_(24)_ = 0.2093, *p* = 0.8359; Figure 2C), indicating that VNS did not affect memory across the 15 min intertrial interval. 

### 3.3. Novel Object Recognition

Next, we tested how acute VNS affects the ability of rats (Sham, *n* = 8; VNS, *n* = 9) to distinguish between a familiar and a novel object as a rodent measure of short-term declarative memory (Figure 3). Rats in the Sham-stimulated and VNS groups showed no differences in overall exploration during the memory test phase (unpaired t-test t_(15)_ = 0.5001, *p* = 0.6242; Figure 3B), suggesting that VNS caused no unspecific changes in locomotion and exploration. On average, rats in both groups also spent similar amounts of time exploring the familiar object (t_(15)_ = 0.243) or the novel object (t_(15)_ = 0.22). However, when the relative time that each animal spent with the familiar and novel object, respectively, was expressed as a recognition index (Figure 3E), it became obvious that rats in the VNS-treated group spent significantly more time investigating the novel object (t_(15)_ = 3.204, *p* = 0.0059). 

### 3.4. Elevated Plus Maze

In both humans [35] and rats [36,37] VNS has been shown to be anxiolytic. A reduction in anxiety could contribute to improved task performance in VNS-treated rats. Therefore, we assessed whether acute VNS reduced anxiety during a single trial in the elevated plus maze (EPM). A separate cohort of rats (*n* = 9 Sham, *n* = 9 VNS) with no previous experience with behavioral tasks or VNS was used for the EPM. Rats were placed in the center of the maze and the time spent in the open and closed arms was analyzed (Figure 4). Unpaired t-tests showed no significant differences between the two groups for time spent in the open arms (unpaired t-test t_(16)_ = 0.3457, *p* = 0.7341; Figure 4B) or for time spent in the closed arms t_(16)_ = 0.8851, *p* = 0.3892; Figure 4C). Similarly, the number of entries into the open (t_(16)_ = 1.531, *p* = 0.1452; Figure 4D) or closed arms (t_(16)_ = 0.2757, *p* = 0.7863; Figure 4E) did not differ between VNS- or Sham-stimulated rats, again suggesting that VNS did not affect locomotion or general exploratory activity. 

## 4. Discussion

VNS has mainly been used to treat clinical disorders such as epilepsy and depression; however, accumulating evidence suggests that VNS might also serve as an effective tool to enhance learning and memory. Vagus nerve stimulation triggers neuromodulator release that mediates cortical plasticity associated with learning. Previous studies have shown that VNS can facilitate learning of sensory and motor behaviors [25,38,39,40], as well as different memory functions [20,21,22,24,26,27,41,42]. Here, we used a series of tasks that require minimal adaptation and previous training to test the acute effects of VNS on cognitive flexibility and short-term memory in naïve rats. 

### 4.1. Rule Shifting

We measured VNS-induced changes in cognitive flexibility via a Cross-maze rule-shifting task. The task requires rodents to acquire a new strategy, while inhibiting the use of a previously reinforced strategy [32,43], task demands that correlate highly with the Reasoning and Problem-solving domain identified by the MATRICS initiative [44]. There is no evidence that stimulation of the left vagus nerve alone is rewarding or reinforcing [23,36] (but see [45]), suggesting that VNS’ effects on cognitive flexibility in the current study were not due to changes in appetitive or motivational properties of the stimuli used in the task. VNS enhanced cognitive flexibility in the CMRST as indicated by the relatively lower number of trials to criterion needed by VNS-treated rats on “Shift-to-Visual-Cue Day”. An analysis of the types of errors committed revealed that VNS-treated rats committed fewer regressive errors and never-reinforced errors, but did not reduce their number of perseverative errors. Perseverative errors are a particularly useful index of how readily animals are able to inhibit the use of the now incorrect strategy. However, a lack of effect of our treatment on this type of error is not unexpected for two reasons: Perseverative errors are a strong indicator of prefrontal cortical dysfunction [46,47], which should not be present in our naïve wildtype rats. In addition, the design of our task, with its division into blocks of 12 trials for analysis, allows only for a comparatively small number of perseverative errors to occur within the range of performance of normal, healthy rats, which based on our experience require about 60–100 trials to reach criterion on the “Shift-to-Visual-Cue Day”; thus, the low baseline number of perseverative errors in healthy Sham-stimulated rats allows for little further improvement by VNS. VNS-treated rats did show a significant reduction in the number of regressive errors, which measure the ability to maintain a novel strategy once perseveration has ceased. Finally, VNS reduced the number of “never-reinforced” errors (responses that are incorrect during both the initial discrimination training and during the shift), which indicates that VNS-treated rats were more efficient in parsing out ineffective strategies [48]. The idea that VNS can enhance cognitive flexibility is also supported by a recent study that looked at the effects of VNS on reversal learning in rats [28].

The ability to inhibit the use of a defunct strategy and enable the learning of a new functional strategy is an important aspect of executive functions, which in rodents are associated with the medial prefrontal cortex [49,50,51] and modulated by monoaminergic and cholinergic afferents [52,53]. Our finding that VNS improved cognitive flexibility is therefore consistent with the so-called neurovisceral integration model which proposes that optimal functioning of prefrontal-subcortical inhibitory circuits is reflected in the vagally-mediated heart rate variability [54]. Higher resting state heart rate variability promotes cognitive flexibility in human subjects [4] and this may explain why activation of the vagus nerve can improve response selection during action cascading [42,55]. However, a recent report also suggests that subdiaphragmatic vagal deafferentation in rats may paradoxically facilitate reversal learning [56]. Combining VNS with the CMRST provides a way to further explore the mechanisms through which vagal afferents can modulate cognitive flexibility. Behavioral rigidity and perseveration occur in a number of neuropsychiatric disorders, including schizophrenia, substance use disorders, obsessive compulsive disorder, and autism [57,58,59,60]. Therapeutic strategies that enhance cognitive flexibility could therefore have important clinical implications. In the next experiments we tested whether the effects of VNS on task performance were specific to the reversal of a previously learned discrimination (i.e., enhanced cognitive flexibility), or whether they influenced learning and memory processes more generally.

### 4.2. Short-Term or Working Memory

Working memory is the delay-dependent representation of stimuli that are used to guide behavior in a task. In rodents, any short-term memory for an object, stimulus, or location that is used within a testing session is often defined as working memory [61,62]. Delayed spontaneous alternation in a T-maze is a measure of short-term spatial memory that capitalizes on the innate tendency of rats and mice to alternate arm choices on repeated trials, which may reflect a tendency of rodents for exploration [63]. Impaired performance in the Spontaneous Alternation task can result from lesions of the medial PFC or corticolimbic pathways that lead to behavioral disinhibition [61]. A previous report suggests that subdiaphragmatic vagal deafferentation does not affect working memory in a nonspatial alternation task [56]. In contrast, we show here that rats which received VNS showed a significant increase in the frequency of alternations, suggesting that VNS can improve working memory, consistent with a previous report in patients with epilepsy [27]. A previous report has shown that, in rats, intertrial delays of 10 min or more result in chance performance in alternations in a Y-maze [64]. Consistent with this idea we found low numbers of alternations across trials separated by 15 min intervals, and this was not improved by VNS, suggesting that these events were not associated in memory and thus not amenable to modulation by acute VNS.

### 4.3. Novel Object Recognition

An important aspect of episodic memory is recognition memory, which in rodents can be assessed via the novel object recognition task. In the task, animals are allowed to examine two or more objects, of which one has been previously investigated by the animal. The relative amount of time taken to explore the new object provides an index of recognition memory. Performance on the task relies heavily on interactions between PFC and hippocampal circuits [65], two brain areas that previously were shown to be strongly modulated by VNS [7,9,11,23,66,67]. A previous report suggests that a loss of vagal forebrain innervation by subdiaphragmatic vagal deafferentation does not affect object recognition memory at various intervals [56]. In contrast, several previous studies, both in rodent models of disease [68,69], as well as in naïve animals [22], have shown that VNS can facilitate episodic memory, supporting earlier findings in patients treated for epilepsy [26,70]. Here we administered VNS acutely during the intertrial interval in an effort to selectively aid consolidation of the memory for the familiar object. Consistent with the idea that acute VNS can enhance short term memory, VNS-treated rats spent relatively more time with the novel object than Sham-treated rats. This is consistent with a previous report in epileptic patients which found that VNS had no effect on learning, but enhanced consolidation, which led to improved retention [70].

### 4.4. Elevated plus Maze

Acute VNS delivery can also produce anxiolytic effects [37,71], and this by itself could improve performance in cognitive tasks. To test if our stimulation parameters similarly reduced anxiety, we tested rats on the EPM which utilizes the predisposition of rodents to enter dark, enclosed spaces (approach) and their unconditioned fear of heights/open spaces (avoidance), respectively, to assess anxiety [72]. The previous reports [37,71] that found anxiolytic effects of VNS used either 1× or 5× 30 s stimulations of the vagus nerve over 15 min immediately before the EPM test. Here we used 3× 30 s stimulation with 3 min intervals, but in our hands, VNS did not significantly reduce avoidance or increase exploration in the EPM. The reason for these differences is not clear but may reflect differences in the timing or total amount of VNS applied. Taken together, our results suggest that acute VNS can improve cognitive performance in naïve rats, even in the absence of anxiolytic effects.

How VNS facilitates learning and memory is an area of active investigation. Stimulation of ascending fibers of the left cervical vagus nerve leads to the release of several neuromodulators, including norepinephrine (NE), acetylcholine, serotonin, and BDNF [7,8,9,10,11,12,13,14,66,67,73], causing rapid widespread cortical and subcortical activation [14,15,16,17,74]. All of these neuromodulators facilitate the induction and/or expression of long-term synaptic plasticity [75,76,77,78], biasing the manner in which cortical networks process information. VNS-induced activation of one or several of these systems may thus modulate synaptic plasticity [79] in response to specific inputs that occur during sensory stimulation [25], or learning and memory [23,38,41,73]. For example, the nucleus of the solitary tract (NTS), which is the prime recipient of ascending vagal projections, regulates NE release from the locus coeruleus (LC) via its projection to the nucleus paragigantocellularis [11,66,80,81,82]. Activation of the NTS → LC pathway and NE release in the hippocampus are important for the consolidation of object-recognition memory [83]. Similarly, activation of the NTS potentiates NE release in the amygdala to enhance retention in emotionally arousing and spatial memory tasks [84,85,86]. In humans, VNS also leads to activation of the LC and improved inhibitory control via noradrenergic mechanisms [87,88]. On the other hand, blocking LC activity reduces VNS’ ability to induce cortical plasticity [73]. 

Autonomic nervous system dysfunction, specifically reduction of parasympathetic vagal activity, is present in mild cognitive impairment, common dementia subtypes, as well as in people with depression [89,90,91,92]. VNS may thus provide an adjunct treatment in a wide variety of disorders in which patients suffer from cognitive impairments. Based on clinical observations in epileptic patients with comorbid disorders [26,70], VNS has been explored as a treatment for depression [93,94,95] and anxiety [35]. A number of recent human studies have furthermore demonstrated the viability of *transcutaneous* or *auricular* VNS to enhance cognitive function [42,87,96,97], which may further increase VNS’ potential in clinical settings. Our findings provide further evidence that even brief VNS can improve cognitive function. Importantly, the tasks that we utilized here are well-characterized and easily implemented, and therefore they can be used in future studies to determine the cellular mechanisms that underlie the behavioral effects described here.

## Figures and Tables

**Figure 1 brainsci-12-01137-f001:**
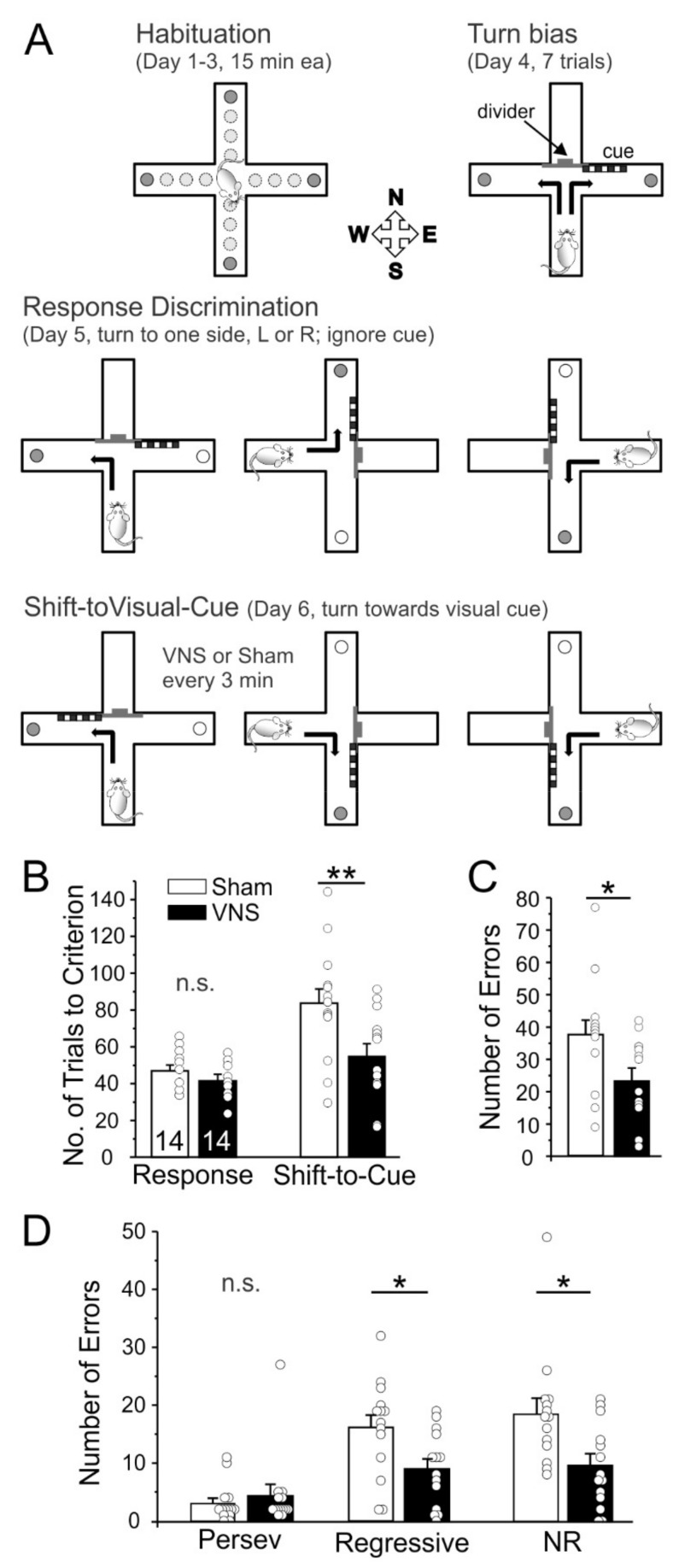
Vagus nerve stimulation (VNS) improves cognitive flexibility. (**A**) Schematic overview of the Cross-Maze Rule Shifting task used to assess cognitive flexibility. Rats were habituated to the maze over 3 days. On the fourth day, animals were tested for an innate preference to turn left or right in the T-maze (Turn bias). On the next day, rats were then trained against their turn bias to learn an egocentric strategy in order to obtain a food reward in one of the two arms of the T-maze (Response Discrimination). Training took place in the presence of a visual cue that had to be ignored at this stage. On the final day, rats were required to shift their strategy, to follow the visual cue to obtain the reward (Shift-to-Visual-Cue). On Shift-to-Visual-Cue Day rats received noncontingent VNS (or Sham-stimulation) every 3 min in one of the stem arms between runs. the (**B**) All animals learned the initial response strategy at the same rate; however, Sham-stimulated rats required a significantly larger number of trials to shift strategies compared to VNS-treated rats. (**C**,**D**) Error analysis based on error types committed during the Shift-to-Visual Cue session. Sham-treated animals committed more total errors, (**C**), and specifically more regressive and never-reinforced (NR) errors, (**D**). Perseverative and Regressive errors are indicative of cognitive flexibility and functions of the medial PFC, while a reduction in NR errors indicates that VNS-treated rats were better at avoiding ineffective strategies and maintaining the effective strategy. Significance is * *p* < 0.05, ** *p* < 0.01, n.s. = non-significant.

**Figure 2 brainsci-12-01137-f002:**
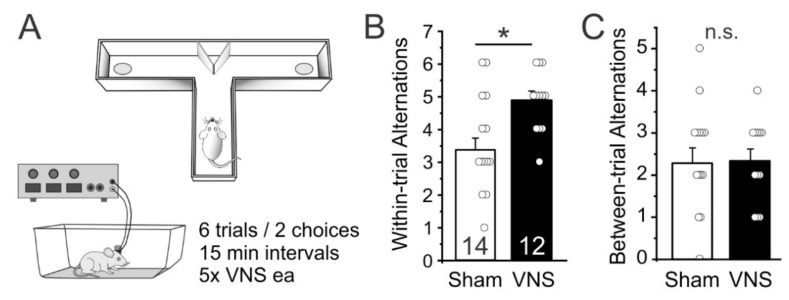
Vagus nerve stimulation (VNS) facilitates foraging behavior and short-term memory. (**A**) Rats were allowed to enter the left or right arm of a modified T-maze. After the choice, animals were confined to that arm for 30 s and then placed back into the stem arm for a second free-choice run. If the animal selected the arm opposite to its initial choice this was considered a spontaneous alternation. Rats performed a total of 6 trials (with 2 choice runs each), with 15 min separating each trial. (**B**) VNS-treated rats showed more spontaneous alternations compared to Sham-stimulated controls. (**C**) VNS did not affect alternation behavior across trials (i.e., the 15 min interval between trials). Significance is * *p* < 0.05, n.s. = non-significant.

**Figure 3 brainsci-12-01137-f003:**
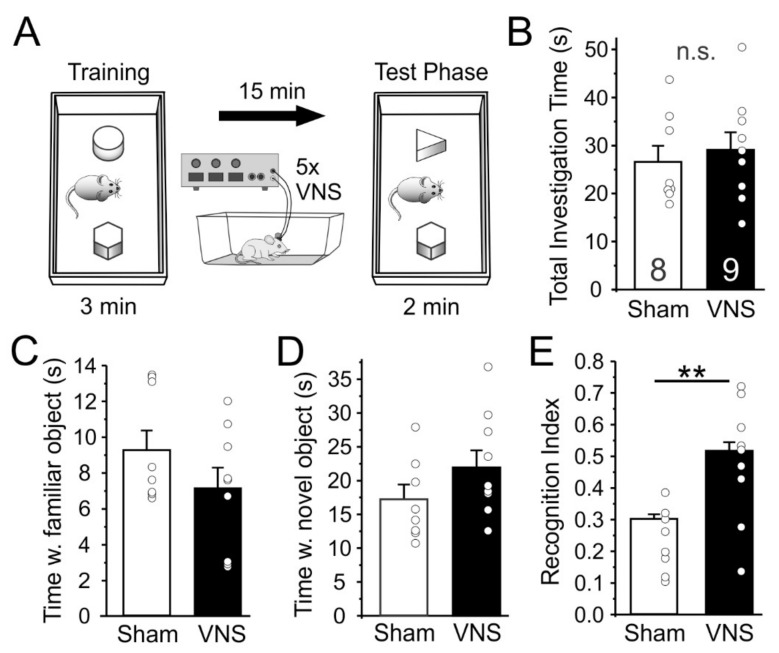
Vagus nerve stimulation (VNS) improves novel object recognition. (**A**) Schematic overview of the setup used to test novel object recognition. Rats were placed in an open box and allowed to investigate two objects for 3 min, after which they were placed back in their home cage for 15 min during which they received VNS or Sham-stimulation every 3 min. One of the objects was replaced with a novel object and animals were then allowed to explore both objects for an additional 2 min. (**B**) Total investigation time during the test phase did not differ between treatment groups, indicating that VNS did not induce unspecific changes in movement and exploration. (**C**,**D**) Total average time exploring the familiar, (**C**), and novel object also did not differ. (**E**) However, exploration time expressed as a recognition index that takes into account the relative time spent with familiar and novel object, respectively, shows that rats in the VNS group spent significantly more time investigating the novel object. Significance is ** *p* < 0.01, n.s. = non-significant.

**Figure 4 brainsci-12-01137-f004:**
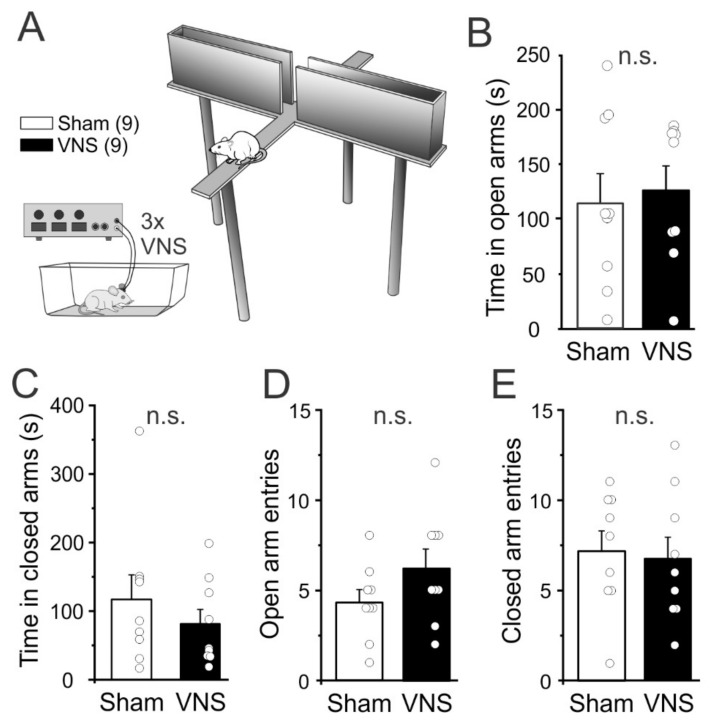
Vagus nerve stimulation (VNS) did not affect measures of anxiety in the elevated plus maze (EPM). (**A**,**B**) Acute VNS (three 30 sec trains, separated by 3 min each, applied immediately before the EPM) did not affect the time spent in the open arms, (**B**), or the closed arms, (**C**), of the EPM. (**D**,**E**) VNS also did not alter the number of entries in the open arms, (**D**), or the closed arms, (**E**). Significance is n.s. = non-significant.

## Data Availability

The data presented in this study are available on request from the corresponding author.

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
