# Peer review of "Acute Vagus Nerve Stimulation Facilitates Short Term Memory and Cognitive Flexibility in Rats"

_brainsci, 2022, doi:10.3390/brainsci12091137_

Round 1
Reviewer 1 Report
The authors describe a series of experiments in which they evaluated the effects of acute electrical vagus nerve stimulation (VNS) on performance in several tasks assessing cognition and affect (strategy shifting, spontaneous alternation, object recognition, and elevated plus maze) in rats. VNS had robust enhancing effects on performance in all three cognitive tasks in comparison to sham VNS, but did not affect measures of anxiety-like behaviour or locomotion in the elevated plus maze. The authors suggest that acute VNS could provide benefits across a variety of cognitive domains.
This is a straightforward and well-designed set of experiments that presents compelling evidence for cognitively-enhancing effects of VNS. The results are, for the most part, clearly explained, and discussed in a scholarly manner. That said, there are a number of methodological points that are unclear (though they do not detract from the overall findings), as well as some comments regarding definitions and relationships of the findings to the existing literature. These are noted below in rough order of importance.
1) It is unclear whether the same or different rats were used in all experiments. Either way, the total number of rats/group should be made explicit in each experiment. In addition, if the same rats were used in all experiments, this prior experience might account for the absence of effects of VNS in the elevated plus maze, as this task may lose sensitivity in animals with significant prior behavioural experience.
2) The statement in the introduction that “less is known about effects of acute VNS (on cognition)” may be reasonable, but it is unclear how “acute” (vs. chronic) is being defined. Earlier studies that assessed VNS effects on memory consolidation (e.g., work by Jensen et al.) used relatively acute stimulation (e.g., a single session of VNS), as have more recent studies that have assessed VNS effects on cognition and affect (e.g., Sun et al. 2017; Sanders et al. 2019; Mathew et al. 2020; Altidor et al. 2021, among others). Given that the issue of a demarcation between acute and chronic may be more semantic than actual, the authors might consider de-emphasizing this point.
3) Compounding the challenges of defining the terms acute vs. chronic, there is insufficient detail in some of the methods to determine exactly how many stimulation trains were delivered. In particular, the description of the shifting task suggests that some rats had hundreds of trials on the shift day, which a) presumably took a long time (hours?), and b) presumably resulted in many VNS trains delivered (in contrast to the other tasks, in which many fewer VNS trains appear to have been delivered). More detail regarding the task duration and number of VNS trains in the shift task would be useful (e.g., was stimulation time-locked to any of the behaviours?)
4) How were the VNS parameters chosen? I believe that with 30 Hz VNS, 0.8 mA is often the optimal amplitude for influencing plasticity and behaviour, although 0.4 mA can exert effects as well.
5) The authors might reconsider the use of a two-factor ANOVA to analyse data in the shift task, as the repeated measure (test day) isn’t really a repeated measure as the tasks were different across the two days. Separate comparisons of the two groups on each of the days would probably be sufficient.
6) There seems to be contradictory information regarding the methods on the object recognition task, as the Methods section says 15 minutes elapsed between the two sessions, and the figure 3 caption says 3 minutes elapsed.
7) The statement in the Discussion that perseverative errors are indicative of PFC dysfunction is accurate, but it does not follow that individuals without PFC dysfunction will not make perseverative errors. As such, this line of argument regarding the absence of VNS effects on this error type does not seem to be that strong.
8) The discussion of heart rate variability (lines 316-318) could be read to imply a causal link between heart rate variability and cognition (greater HRV causes better cognition), and further suggest that VNS is mirroring this causal effect. Is this interpretation accurate? If not, the authors may consider rephrasing this portion of the discussion.
9) On line 104 of the manuscript, what counts as having “passed” the probe trial during training on the shift task?
10) It would be useful to report data on the actual exploration times in the object recognition task, in addition to the ratios.
11) There is a sentence fragment at the end of the Delayed Spontaneous Alternation methods paragraph.
Author Response
Dear Mrs. Zhang,
We would like to thank you and the reviewers for the evaluation of our manuscript "Acute vagus nerve stimulation facilitates short term memory and cognitive flexibility in rat" and the very helpful comments. We have tried to address the reviewers’ concerns as outlined below, and we think that the changes to the manuscript have further improved the manuscript. Thus, we appreciate the opportunity to resubmit this paper for consideration in Brain Sciences. In the manuscript all changes are indicated in red. The major changes include a complete redrawing of the figures to clearly indicate the number of rats used and to incorporate individual data points. As outlined below in detail, we have also corrected a number of very unfortunate mistakes and omissions that occurred in the first version of the manuscript. These pertain to the description of the data analysis (but not the analysis itself), as well as the labeling of some graphs. We are particularly thankful to the reviewers for pointing these mistakes out.
Reviewer 1
The authors describe a series of experiments in which they evaluated the effects of acute electrical vagus nerve stimulation (VNS) on performance in several tasks assessing cognition and affect (strategy shifting, spontaneous alternation, object recognition, and elevated plus maze) in rats. VNS had robust enhancing effects on performance in all three cognitive tasks in comparison to sham VNS, but did not affect measures of anxiety-like behaviour or locomotion in the elevated plus maze. The authors suggest that acute VNS could provide benefits across a variety of cognitive domains.
This is a straightforward and well-designed set of experiments that presents compelling evidence for cognitively-enhancing effects of VNS. The results are, for the most part, clearly explained, and discussed in a scholarly manner. That said, there are a number of methodological points that are unclear (though they do not detract from the overall findings), as well as some comments regarding definitions and relationships of the findings to the existing literature. These are noted below in rough order of importance.
1) It is unclear whether the same or different rats were used in all experiments. Either way, the total number of rats/group should be made explicit in each experiment. In addition, if the same rats were used in all experiments, this prior experience might account for the absence of effects of VNS in the elevated plus maze, as this task may lose sensitivity in animals with significant prior behavioural experience.
We used 47 rats for the 3 cognitive tasks (rule-shifting, spontaneous delayed alternation, and novel object recognition), with most of the animals participating in 2 tasks. However, importantly, a separate cohort of rats (n = 9 Sham, n = 9 VNS), which did not participate in any other task, was used for the elevated plus maze test to avoid the potential effects of previous behavioral experience or exposure to VNS that the reviewer alludes to. This is now better described in the methods and numbers of animals for each test are now clearly indicated in text and figures.
2) The statement in the introduction that “less is known about effects of acute VNS (on cognition)” may be reasonable, but it is unclear how “acute” (vs. chronic) is being defined. Earlier studies that assessed VNS effects on memory consolidation (e.g., work by Jensen et al.) used relatively acute stimulation (e.g., a single session of VNS), as have more recent studies that have assessed VNS effects on cognition and affect (e.g., Sun et al. 2017; Sanders et al. 2019; Mathew et al. 2020; Altidor et al. 2021, among others). Given that the issue of a demarcation between acute and chronic may be more semantic than actual, the authors might consider de-emphasizing this point.
The reviewer’s point is well taken. Because we have no way of delineating acute vs. chronic stimulation (even in our own experiments) we have de-emphasized this distinction in the introduction as suggested.
3) Compounding the challenges of defining the terms acute vs. chronic, there is insufficient detail in some of the methods to determine exactly how many stimulation trains were delivered. In particular, the description of the shifting task suggests that some rats had hundreds of trials on the shift day, which a) presumably took a long time (hours?), and b) presumably resulted in many VNS trains delivered (in contrast to the other tasks, in which many fewer VNS trains appear to have been delivered). More detail regarding the task duration and number of VNS trains in the shift task would be useful (e.g., was stimulation time-locked to any of the behaviours?)
Unfortunately, we were not able to deliver VNS contingently with any behaviors (e.g., correct choices in the rule-shift task). Instead, as outlined in the Methods (lines 81-82) we applied non-contingent VNS every 3 minutes. As the reviewer correctly implies, the number of stimulations given depended on the subject’s performance; however, no rat took longer than 120 minutes to complete the task, thus we can estimate a maximum of 40 VNS applications.
4) How were the VNS parameters chosen? I believe that with 30 Hz VNS, 0.8 mA is often the optimal amplitude for influencing plasticity and behaviour, although 0.4 mA can exert effects as well.
We chose these parameters based on our previous work where we found both behavioral effects on extinction learning, as well as changes in synaptic plasticity (Childs et al., 2017; Childs et al., 2019; Pena et al., 2014). We have added an explanation and references in the Methods section (lines 87-88).
5) The authors might reconsider the use of a two-factor ANOVA to analyse data in the shift task, as the repeated measure (test day) isn’t really a repeated measure as the tasks were different across the two days. Separate comparisons of the two groups on each of the days would probably be sufficient.
We chose a 2-factor ANOVA precisely to account for the change in task. Because the task changes there should be a main effect of “test day”. If there was not (i.e., if we had to accept the null hypothesis that there is no change) then any difference between the treatment groups on shift day would be more difficult to interpret.
6) There seems to be contradictory information regarding the methods on the object recognition task, as the Methods section says 15 minutes elapsed between the two sessions, and the figure 3 caption says 3 minutes elapsed.
This is one of several regrettable mistakes we need to apologize for. The labelling in the figure was wrong. The interval was indeed 15 min to allow for more than a single VNS stimulation. This has been corrected in the figure.
7) The statement in the Discussion that perseverative errors are indicative of PFC dysfunction is accurate, but it does not follow that individuals without PFC dysfunction will not make perseverative errors. As such, this line of argument regarding the absence of VNS effects on this error type does not seem to be that strong.
The reviewer is correct that individuals with PFC dysfunction (or wildtype rats as in our case) also make perseverative errors; however, they do so less frequently. In addition, few perseverative errors should be expected in our task due to the way how we operationalized these errors in our task (line 119-123), and due to this floor effect there is thus little to improve upon by VNS. We have rewritten this sentence in an attempt to make this limitation of our analysis clearer.
8) The discussion of heart rate variability (lines 316-318) could be read to imply a causal link between heart rate variability and cognition (greater HRV causes better cognition), and further suggest that VNS is mirroring this causal effect. Is this interpretation accurate? If not, the authors may consider rephrasing this portion of the discussion.
The neurovisceral integration model by Thayer and Lane (Thayer and Lane, 2000) indeed proposes that cardiac vagal tone, indexed via heart rate variability, can indicate the functional integrity of neural networks that control emotion–cognition interactions. Individual differences in heart rate variability are related to performance on tasks associated with executive function and prefrontal cortical activity. Higher resting heart rate variability is associated with more adaptive and functional top-down and bottom-up cognitive modulation. Manipulating resting heart rate variability levels is associated with changes in performance on executive-function tasks.
9) On line 104 of the manuscript, what counts as having “passed” the probe trial during training on the shift task?
The probe trial requires the subject to follow the same “new” rule, i.e., to turn towards the visual clue; however, the trial starts from the previously unused “North” arm in order to prevent rats to use an allocentric response strategy. We have this further clarified in the Methods section (lines 116-118).
10) It would be useful to report data on the actual exploration times in the object recognition task, in addition to the ratios.
We have added the data for actual investigation times to Figure 3 and we have corrected the erroneous labelling of the Y-axis in Figure 3B.
11) There is a sentence fragment at the end of the Delayed Spontaneous Alternation methods paragraph.
Thank you very much for pointing this out, this happened probably when copying the text into the Brain Sciences template. The full sentence should read: “In addition, we also analyzed alternations across the 15 min intertrial interval in an effort to assess potential changes in memory on a longer time scale.”
Reviewer 2 Report
Summary: The authors tested the effects of acute vagus nerve stimulation on short term memory and cognitive flexibility in healthy rats with several different behavioral tasks. Their results suggest a benefit of acute VNS on short-term memory, but not on locomotion or anxiety. These are an interesting, if narrow, set of results with broader future implications for use in human patients. There are moderate concerns regarding the methodological approaches, data description and data analysis. Of key importance, the authors used an incorrect statistical analysis (paired t-test for unpaired data).
Stimulation parameters: The specific stimulation parameters applied are not justified through testing or prior literature. Please explain the rationale for these parameters.
Relatedly, the stimulation parameters used in experiments differ from those used during surgical procedures to test the device. Why were different parameters chosen during surgery, and do the experimental parameters still cause a cessation of breathing?
Can you provide evidence to demonstrate that the shorted cuffs did not apply stimulation (ie no change in breathing rate)?
Did awake animals receiving stimulation exhibit signs of perception of the stimulus?
Experimental design: First, the experimental design for each test is somewhat complex, and a schematic for the experimental timeline for each test would be very helpful.
All of the tests, and the evaluation metrics need to be described with much more detail. Examples include:
· Fig 1. Define the outcomes perseverative, regressive and never reinforced in much greater detail and with an emphasis on the behavioral outcomes. How were each of these outcomes quantified?
· Figure 1B: what is the criterion for shift-to-cue? Is it the same criterion as response phase?
· Fig 2. What is between-trials alteration and how is it quantified?
Please justify why animals received VNS during the habituation phase. Are there concerns that this may have influenced the subsequent results?
Statistical design: Please report the number of animals for each panel of each figure, and when applicable, the number of trials.
If there is a comparison between groups, an unpaired t-test should be applied (if the normality test has been passed). Paired t-tests are not appropriate for this data.
Please confirm that the data was calculated on a per-animal basis, rather than a per-trial basis. In fact, we recommend that you include dots to reflect each individual data point in addition to the summary bars on the bar plots.
Figure 1: Please quantify the number of animals that were excluded because they couldn’t learn the task within 200 trials.
Figure 4: It seemed like VNS did spend less time in closed arms and slightly more time in open arm. Did the author test the ratio of time in open arm to the time in close arm (or overall time)? Also, please provide the p-values for the statistics in this figure.
Please mention panel E in the legend.
Minor:
Line 119: Please provide reference for this task.
Line 130, 222: The last sentence is unfinished.
Line 280: Please also cite Bowles, Hickman, Peng et al., 2022, Neuron
Additional proof-reading is required to fix spelling errors, extra spaces, etc.
Author Response
Dear Mrs. Zhang,
We would like to thank you and the reviewers for the evaluation of our manuscript "Acute vagus nerve stimulation facilitates short term memory and cognitive flexibility in rat" and the very helpful comments. We have tried to address the reviewers’ concerns as outlined below, and we think that the changes to the manuscript have further improved the manuscript. Thus, we appreciate the opportunity to resubmit this paper for consideration in Brain Sciences. In the manuscript all changes are indicated in red. The major changes include a complete redrawing of the figures to clearly indicate the number of rats used and to incorporate individual data points. As outlined below in detail, we have also corrected a number of very unfortunate mistakes and omissions that occurred in the first version of the manuscript. These pertain to the description of the data analysis (but not the analysis itself), as well as the labeling of some graphs. We are particularly thankful to the reviewers for pointing these mistakes out.
Reviewer 2
Summary: The authors tested the effects of acute vagus nerve stimulation on short term memory and cognitive flexibility in healthy rats with several different behavioral tasks. Their results suggest a benefit of acute VNS on short-term memory, but not on locomotion or anxiety. These are an interesting, if narrow, set of results with broader future implications for use in human patients. There are moderate concerns regarding the methodological approaches, data description and data analysis. Of key importance, the authors used an incorrect statistical analysis (paired t-test for unpaired data).
Stimulation parameters: The specific stimulation parameters applied are not justified through testing or prior literature. Please explain the rationale for these parameters.
As outlined in response to point #4 by reviewer 1 we chose these parameters based on our previous work where we found both behavioral effects on extinction learning, as well as changes in synaptic plasticity (Childs et al., 2017; Childs et al., 2019; Pena et al., 2014). We have added an explanation and references in the Methods section (lines 87-88).
Relatedly, the stimulation parameters used in experiments differ from those used during surgical procedures to test the device. Why were different parameters chosen during surgery, and do the experimental parameters still cause a cessation of breathing?
The stimulation parameters that we use during surgery are sufficient to cause a cessation of breathing during anesthesia (even though we use lower stimulation amplitudes than during the behavioral tests), which can be used as a quick test of the functionality of the VNS cuffs (Butler et al., 2021). During normal breathing in awake animals the stimulation parameters do no recruit the Hering-Breuer reflex.
Can you provide evidence to demonstrate that the shorted cuffs did not apply stimulation (ie no change in breathing rate)?
Yes, sham-stimulated rats never show evidence of the Hering-Breuer reflex.
Did awake animals receiving stimulation exhibit signs of perception of the stimulus?
Rats that receive VNS using these stimulation parameters do not interrupt ongoing behavior. For example, we have previously shown that VNS with these parameters show no changes in appetitive behavior, e.g., responding for food rewards (Childs et al., 2017). However, rats need to be habituated to being tethered at the headcap and at the time we also habituated rats to any possible sensation from VNS. Importantly, this happened several days before the actual test day, so it is unlikely that the VNS influenced learning and memory at the time.
Experimental design: First, the experimental design for each test is somewhat complex, and a schematic for the experimental timeline for each test would be very helpful.
We have added additional information to the figures, which hopefully with the revised methods and figure legends will make it easier to envision how the experiments were performed.
All of the tests, and the evaluation metrics need to be described with much more detail. Examples include:
- Fig 1. Define the outcomes perseverative, regressive and never reinforced in much greater detail and with an emphasis on the behavioral outcomes. How were each of these outcomes quantified?
This is another unfortunate omission in the previous version of the manuscript. We have now added functional definitions of the error types and explain how they were measured into the Methods section (lines 131-142). - Figure 1B: what is the criterion for shift-to-cue? Is it the same criterion as response phase?
Yes, the criterion is the same, 9 out of 10 consecutive responses had to be correct to reach criterion. We had stated in the Methods that “The training and response criteria for the Shift-to-Visual-Cue Discrimination were identical to those during Response Discrimination” (lines 124-125) - Fig 2. What is between-trials alteration and how is it quantified?
The description of “between-trials alternation” also unfortunately was deleted when we copied our text into the Brain Sciences template. We apologize for that. The description should read “In addition, we also analyzed alternations across the 15 min intertrial interval in an effort to assess potential changes in memory on a longer time scale” This has now been added to the Methods section again (lines 154-156).
Please justify why animals received VNS during the habituation phase. Are there concerns that this may have influenced the subsequent results?
As outlined above, the stimulation parameters that we use for VNS do not noticeably disrupt behavior; however, it is possible that rats perceive the VNS. More importantly though, rats needed to be habituated to the tether through which VNS or Sham stimulation was delivered. In order to minimize both of this potential confounds for our cognitive tasks we habituated rats to the tether and also applied brief VNS or sham-stimulation every 3 minutes over 15 minutes for 3 days.
Statistical design: Please report the number of animals for each panel of each figure, and when applicable, the number of trials.
We have added the number of animals to each figure. We have also added individual data points for each animal to the figures to provide a better idea of the variability.
If there is a comparison between groups, an unpaired t-test should be applied (if the normality test has been passed). Paired t-tests are not appropriate for this data.
This was an unfortunate and very regrettable error in the description of the Methods. All tests indeed used unpaired t-tests for two-samples with equal variance. We also reran all stats to confirm this.
Please confirm that the data was calculated on a per-animal basis, rather than a per-trial basis. In fact, we recommend that you include dots to reflect each individual data point in addition to the summary bars on the bar plots.
Data reflects the average values for each subject. Individual data points have now been added to all figures.
Figure 1: Please quantify the number of animals that were excluded because they couldn’t learn the task within 200 trials.
No animals were excluded from analysis because they did not reach criterion on Shift-to-Cue Day after 200 trials. The 200-trial cut-off number is a (very generous) number that we defined a-priori based on extensive experience with this task. Because this did not apply to any rats in this study, we deleted this description to avoid confusion.
Figure 4: It seemed like VNS did spend less time in closed arms and slightly more time in open arm. Did the author test the ratio of time in open arm to the time in close arm (or overall time)? Also, please provide the p-values for the statistics in this figure.
Please mention panel E in the legend.
We have included individual data points in the figures (as in all figures) and we have now correctly labeled the figure legend. We have added an indicator of significance (or lack thereof) into the figure. We did calculate a ratio of time in the open arms relative to time spent in the closed arms; however, this ratio did not indicate that acute VNS significantly altered behavior in the EPM.
Minor:
Line 119: Please provide reference for this task.
A reference has been added
Line 130, 222: The last sentence is unfinished.
This has been corrected
Line 280: Please also cite Bowles, Hickman, Peng et al., 2022, Neuron
This reference has been added
Additional proof-reading is required to fix spelling errors, extra spaces, etc.
We believe that all these errors and typos have been corrected now.
Round 2
Reviewer 2 Report
The revisions provided by the authors have strengthened the manuscript and addressed concerns. The new figures with schematics and additional data are more compelling.